# Simulating Landfast Ice in Lake Superior

Yuchun Lin [1,*], Ayumi Fujisaki-Manome [2,3] and Eric J. Anderson [4,5]

1    Center of Excellence for Ocean Engineering, National Taiwan Ocean University, Keelung City 20224, Taiwan
2    Cooperative Institute for Great Lakes Research, University of Michigan, Ann Arbor, MI 48108, USA;
     ayumif@umich.edu
3    Climate & Space Sciences and Engineering, University of Michigan, Ann Arbor, MI 48104, USA
4    Civil & Environmental Engineering, Colorado School of Mines, Golden, CO 80401, USA;
     ejanderson@mines.edu
5    Hydrologic Science and Engineering, Colorado School of Mines, Golden, CO 80401, USA
*    Correspondence: yuchlin@mail.ntou.edu.tw

**Abstract:** Landfast ice plays an important role in the nearshore hydrodynamics of large lakes, such as the dampening of surface waves and currents. In this study, previously developed landfast ice basal stress parameterizations were added to an unstructured grid hydrodynamic ice model to represent the effects of grounded ice keels and tensile strength of ice cover. Numerical experiments using this model were conducted to evaluate the development of coastal landfast ice in Lake Superior. A sensitivity study of the free parameters was conducted from December 2018 to May 2021 to cover both high and low ice cover winters in Lake Superior and was compared against observations from the United States National Ice Center. The model reproduces the annual variation in coastal landfast ice in Lake Superior, particularly in shallow nearshore areas and the semi-enclosed bays in the northern regions of the lake. Experiments also show that the growth of landfast ice is mainly controlled by the free parameter that controls the critical ice thickness for the activation of basal stress. Overall, the model tends to underestimate the extent of coastal landfast against observations.

**Keywords:** landfast ice; basal stress; numerical simulation; Great Lakes; lake ice; ice modeling





## 1. Introduction

Landfast ice (or fast ice) is ice cover that is immobile near coastal regions. In the Arctic, the majority of landfast ice is often found in shallow coastal waters from the beginning of fall and into spring [1,2]. In some regions of the Arctic, landfast ice can last until early summer when the warm temperatures return to the region [3]. Landfast ice coverage can extend offshore into the ocean for a few tens of km, e.g., in the Chukchi and Beaufort Seas, and hundreds of km, e.g., along the coast of Siberia where the landfast ice is the most prominent [4–6]. Long-term analysis of observations showed that the Arctic landfast ice extent was significantly decreased in 1979 to 2018 and it might disappear during summer in the near future [7].

In general, landfast ice affects the interactions between the atmosphere and sea water or freshwater in many aspects, including the transfer of heat, moisture, and momentum flux. For example, the interactions between the landfast ice, cold winds, and the coast can quickly decouple the onshore area and form a thinner ice layer at the surface, which helps determine the offshore positions of polynyas [5]. Furthermore, observations show that the landfast ice can affect the deeper Arctic Ocean, e.g., the formation of the halocline layer [8] and brine formation [9–12]. Landfast ice is also found to be important in simulating sea surface height [13] and ice thickness [14].

Numerical approaches have been established to simulate and understand landfast ice over the Arctic. Specifically, tensile strength was added to simulate landfast ice [8,15]. Òlason et al. [16] adjusted mechanical parameters in a viscous-plastic sea ice model with high spatial resolution for modeling landfast ice in the Kara Sea. Ice drifting velocity

was set to zero to capture landfast ice for certain regions in both the model [14] and observations [17]. Lieser [18] included the condition of thickness over a defined water depth in addition to the zero ice velocity criterion. Wang et al. [19] developed an ice-ocean model along the coast of Alaska and defined landfast ice as ice cover in water shallower than 35 m and with a velocity smaller than 4 cm s$^{-1}$. The same criteria are applied to the basal stress parameterization for landfast ice in the Arctic [20,21].

Almost all the above-mentioned studies were focused on coastal regions of the Arctic. However, landfast ice is also common in large freshwater lakes with seasonal ice cover, such as the North American Great Lakes (hereafter the Great Lakes). Despite several distinctions from the ocean environment (e.g., absence of brine, short fetch), landfast ice cover also plays a significant role in nearshore lake hydrodynamics. Furthermore, in the Great Lakes, landfast ice in the major waterways and ports poses a challenge for navigation and icebreaking operations that assist commercial shipping. Several numerical model applications to predict Great Lakes ice cover have been presented [22–30], including the model used for the National Oceanic and Atmospheric Administration (NOAA)'s Great Lakes Operational Forecast System (GLOFS) [29,31]. However, these applications did not include a specific parameterization to represent or evaluate the models' abilities to simulate landfast ice. In addition, the parameterizations to represent landfast ice (i.e., the basal stress and the ice tensile strength) were never tested in the unstructured grid hydrodynamic ice models.

In this study, we address these gaps by incorporating the basal stress parameterization [20] and the ice tensile strength parameterization into the Finite Volume Community Ocean Model (FVCOM) [32,33], which is coupled with the unstructured grid version of the Los Alamos Sea Ice Model (UG-CICE) [34]. We apply the FVCOM-CICE model to Lake Superior, conduct a sensitivity study by perturbing the free parameters within the landfast ice parameterizations, and examine the modeled development of coastal landfast ice in Lake Superior. The parameterization of landfast ice and the observations are described in Section 2; the numerical results are presented in Section 3; and the results are summarized in Section 4.

## 2. Materials and Methods

The unstructured Finite Volume Community Ocean Model (FVCOM) [32,33] is applied to Lake Superior in this study. Figure 1a,b show the bathymetry and the unstructured FVCOM mesh grid, respectively, of Lake Superior. FVCOM is a three-dimensional, unstructured, free-surface, primitive equation, sigma-coordinate oceanographic model that solves the integral form of the governing equations. FVCOM has been applied in several studies of the coastal ocean, including successful application to operational forecasting in the Great Lakes [29,30,34–38]. Horizontal grid resolution in the model ranges from roughly 200 m near the shoreline to 2500 m offshore, with 21 vertical sigma layers evenly distributed throughout the water column. As a result, the FVCOM model for Lake Superior contains 174,015 elements. Modeled depths are taken from 3 arc-second bathymetry data from the NOAA National Centers for Environmental Information (NCEI).

FVCOM is forced by surface meteorological data from the High-Resolution Rapid Refresh (HRRR) [39,40]. HRRR is a 3 km data-assimilated implementation of the Weather Research and Forecasting model and updated hourly for operational weather forecasting by the NOAA National Centers for Environmental Prediction (NCEP). FVCOM is coupled with the unstructured grid version of the Los Alamos Sea Ice Model (UG-CICE) [34]. UG-CICE is based on a previous version of the Los Alamos Sea Ice Model (CICE, CICE consortium, 2022), whose latest version (version 6) includes the parameterizations for the basal stress and the ice tensile strength. At the time of the work by Gao et al. [34], these parameterizations were not incorporated in UG-CICE. In this work, following Lemieux et al. [20,21], we incorporate these parameterizations into the UG-CICE model that has been applied to the Great Lakes [29,30,41].

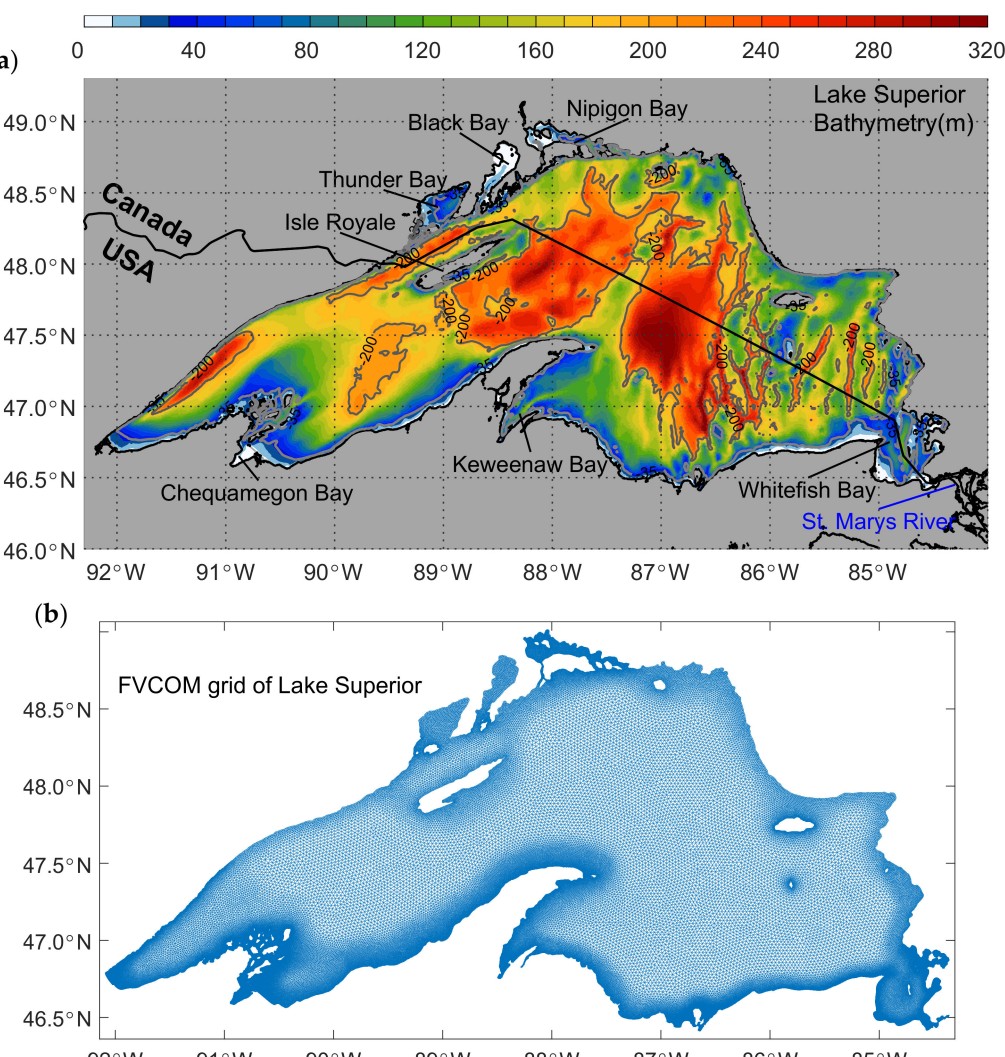

**Figure 1.** (**a**) Bathymetry (meter) of Lake Superior of the FVCOM. The two contours represent the isobaths of 35 m and 200 m. The 35 m isobath represents the criterion of depth for determining landfast ice in Lake Superior, and the 200 m isobath represents the deeper lake. Larger bays, including Nipigon Bay, Black Bay, Thunder Bay, Chequamegon Bay, Keweenaw Bay, and Whitefish Bay, Isle Royale, and the St. Marys River, which connects Lake Huron, are marked. (**b**) FVCOM mesh grid of Lake Superior.

The dynamics of sea ice or lake ice are mainly in the vertical direction and the processes are often considered a two-dimensional problem [42]. Lemieux et al. [20] developed a simple parameterization using the basal stress term $\tau_b$ to represent the subgrid scale grounded ice in the momentum equation in the vertical direction. A critical mean thickness $h_c$ is defined to represent the grounded ice (or landfast ice). For the case of mean ice thickness greater than the critical mean ice thickness, the ice is grounded at the bottom and the hydrostatic balance is broken. As the ice is grounded, a normal force pointed upward appears, which is equal to the weight of the ridge minus the weight of the displaced water. Ref. [20] defined the basal stress $\tau_b$ that represents the above-mentioned process and was incorporated into the momentum equation of ice:

$$\tau_b = \begin{cases} 0 \\ k_2 \left( \frac{u}{|\mathbf{u}| + u_0} \right) (h - h_c) exp^{-C_b(1-A)} \text{, if } h > h_c \end{cases} \tag{1}$$

where the $k_2$ is a free parameter with units of $\text{Nm}^{-3}$, $u$ is the velocity component in $x$-axis, $|\mathbf{u}|$ is the speed of drifting ice, $u_0$ is a small velocity parameter for a smooth transition between the static and kinetic friction regimes, $h$ is the mean ice thickness in a grid cell, $h_c = Ah_w/k_1$ is the critical mean thickness, $h_w$ is the water depth, and $A$ is the ice concentration (0–1). $k_1$ is the first free parameter and is also called the critical thickness parameter, and $k_2$ is called the maximum basal stress parameter [20]. The maximum basal stress that can be sustained by the grounded ridge depends on the weight of the ridge in excess of the hydrostatic balance. The theoretical estimates by Lemieux et al. [20] suggest that $k_1$ should be around 10 and $k_2$ should be greater than $5 \text{ Nm}^{-3}$. They also note that $k_2$ is highly sensitive to the areal fraction of large ridges in an ice-covered area, as well as the mean ice thickness. Their sensitivity tests of the Arctic Ocean with 25 km horizontal resolution presented the best results with $k_1 = 8$ and $k_2 = 15$. However, they note that $k_1$ and $k_2$ should be optimized through numerical experiments.

In their follow-on study, Lemieux et al. [21] examined the impacts of ice tensile strength on modeled landfast ice coverage. Isotropic tensile strength was introduced by updating the parameterization of the bulk viscous coefficient $\zeta$ in the governing equations for ice internal stress to the following:

$$\zeta = P_p(1 + T)/2\Delta^* \tag{2}$$

where $P_p$ is the compressive ice strength [43], $\Delta^* = (\Delta + \Delta_{min})$, $\Delta$ is a function of ice divergence, the horizontal tension rate, the shearing strain rate, and the ratio of major and minor axes of the elliptical yield curve. $\Delta_{min}$ is a small value added to avoid a singularity with $\zeta$. $T$ is the isotropic tensile strength parameter and characterizes the amount of tensile strength as a function of the ice strength in compression ($TP_p$), and $P = P_p\Delta/\Delta^*$ is a replacement pressure to ensure the internal stresses are zero when the strain rates are zero. Details on the governing equations for ice internal stress, implementation of the isotropic tensile strength parameter, and its impacts on elliptical yield curves are detailed in Lemieux et al. [21] and are thus omitted in this paper. Lemieux et al. [21] examined $T = 0$, 0.1, 0.2, 0.3, and 0.4 for their application to the Arctic Ocean and found that $T$ should be between 0.1 and 0.2 to obtain the most realistic results.

In this study, the control case is the simulation without the basal stress effect. The values of $k_1$ are changed between 8, 16, and 48, the values of $k_2$ vary between 15 and $60 \text{ Nm}^{-3}$, and for the isotropic tensile strength parameter $T$, 0, 0.2, and 0.4 were tested. These values are selected based on the findings of Lemieux et al. [20] except for $k_1 = 48$, which is used to examine an extreme case. Using the modeled ice output, landfast ice is defined by ice concentrations greater than 90%, which is the first criterion. The second criterion for determining the landfast ice in the model is the water depth, which is set to be smaller than 35 m following the definition of Lemieux et al. [20,21]. Finally, the third criterion for landfast ice is the ice drifting speed. We have examined several values for the speed criterion and chosen a threshold of $0.4 \text{ cm s}^{-1}$ to represent the landfast ice in the model, which is ten times smaller than the criterion applied to the marginal seas of the Arctic. This threshold also restricts the overestimated landfast ice in the control case. Modeled landfast ice does not change significantly when the threshold varies from $0.2 \text{ cm s}^{-1}$ to $4 \text{ cm s}^{-1}$. In addition, the ice drifting threshold of the Arctic was set up according to the mean ice drifting speed in the winter [20], which may be inappropriate for Lake Superior.

In our application, there are several distinct features that are different from the previous studies focused on the Arctic [20,21]. These features include the unstructured grid configuration with much higher resolution nearshore (~200 m or higher) and thinner ice thickness (~20 cm in mid-winter) in Lake Superior. A set of sensitivity tests similar to [20,21] are conducted to examine the free parameters ($k_1$, $k_2$, and $T$).

Observational data of Great Lakes ice was obtained from the U.S. National Ice Center (NIC; [44]) for validating the modeled ice. Through a bi-national coordinated effort between the U.S. NIC and Canadian Ice Center, routine ice analysis products are produced from available data sources including Radarsat-2, Envisat, AVHRR, Geostationary Operational

and Environmental Satellites (GOES), and Moderate Resolution Imaging Spectroradiometer (MODIS). The NIC provides shapefiles and gridded ascii files that include the information for multiple ice types, including landfast ice. The shapefile contains polygons with information on minimum, mid-range, and maximum total ice concentration, and the concentration of multi-year, first-year ice, and ice types. According to the user guide of NIC, the Sea Ice Grid (SIGRID) codes represent ice concentrations from 10% to 90% with 10% intervals, e.g., SIGRID = 10 for 10%, and 90 for 90%. The SIGRID code is 91 for ice concentration higher than 90% and is 92 for landfast ice. The gridded ascii files contain ice codes representing different types of ice, e.g., ice concentration and landfast ice, on a spatial resolution of 2.5 km $\times$ 2.5 km. For the simulation period in this study, we confirmed that the landfast ice coverage timeseries are almost identical between the shapefile and gridded ascii products. Therefore, we used the gridded ice data product for the post-simulation analysis largely due to the simplicity of the data processing. The time period investigated is from 20 December 2018 to 30 May 2021 to cover anomalously cold (2018/2019) and warm (2020/2021) winters.

## 3. Results and Discussion

### 3.1. Observed Landfast Ice

Figure 2a shows the timeseries of landfast ice area from NIC of Lake Superior between December 2018 to May 2021. The landfast ice in 2018/19 is much higher than the other two years due to the severe cold winter affected by the polar vortex. In general, the date of maximum landfast ice area is around late February and early March. In these three years, the maximum areas of landfast ice are approximately 11,589 km$^2$ on 27 February 2019, 264 km$^2$ on 3 March 2020, and 5530 km$^2$ on 20 February 2021. The previously used criteria to determine landfast ice are that the ice drifting speed is less than 4 cm s$^{-1}$ and the ice is located at a water depth less than 35 m [19–21]. The ice drifting speed is not available from NIC. Thus, in order to evaluate landfast ice, we used the water depth criteria for the observed landfast ice from NIC. In Figure 2a, the gray and the black colors represent landfast ice extent for water depths greater and smaller than 35 m, respectively. A large portion of landfast ice is observed at a water depth greater than 35 m in the winter of 2018/19. On average, landfast ice located at depths greater than 35 m accounts for about 25% of coverage in the winter of 2018/19. For the winters of 2019/20 and 2020/21, the value drops to ~9% and ~12%, respectively. Note that the presence of landfast ice located at greater water depths generally occurred during high ice periods in late February and early March. However, observed landfast ice is subject to some level of uncertainty because satellite observations used by the NIC are limited by cloud cover for visible sensors and by spatial footprints for synthetic aperture radars (SARs). Furthermore, these satellite measurements do not directly provide ice drifting speed to determine whether ice is immobile or mobile. Therefore, whether ice cover is reported as landfast or not often involves subjective judgement. Such uncertainty may be greater for landfast ice reported in the deeper areas, where grounding of ice ridges does not play a role, and immobility is solely driven by internal stress.

Figure 2b shows a snapshot of the maximum coverage of landfast ice on 27 February 2019, which is represents the largest extent during the study period. It is obvious that the coverage of landfast ice determined by the NIC extended to depths greater than 35 m. The landfast ice reaches the area beyond 200 m depth in the strait north of Isle Royale, the middle of Whitefish Bay, and western Lake Superior. On 27 February 2019, the portion of landfast ice located at water depths greater than 35 m reaches 46% of the total landfast ice area. Similar spatial patterns are often observed in the other winters when the ice cover in Lake Superior is high. Daily snapshots show that landfast ice initially forms in the two bays in the north, Black Bay and Nipigon Bay, followed by other shallower regions, e.g., Whitefish Bay in the southeast of Lake Superior or Chequamegon Bay in the southwest (not shown).

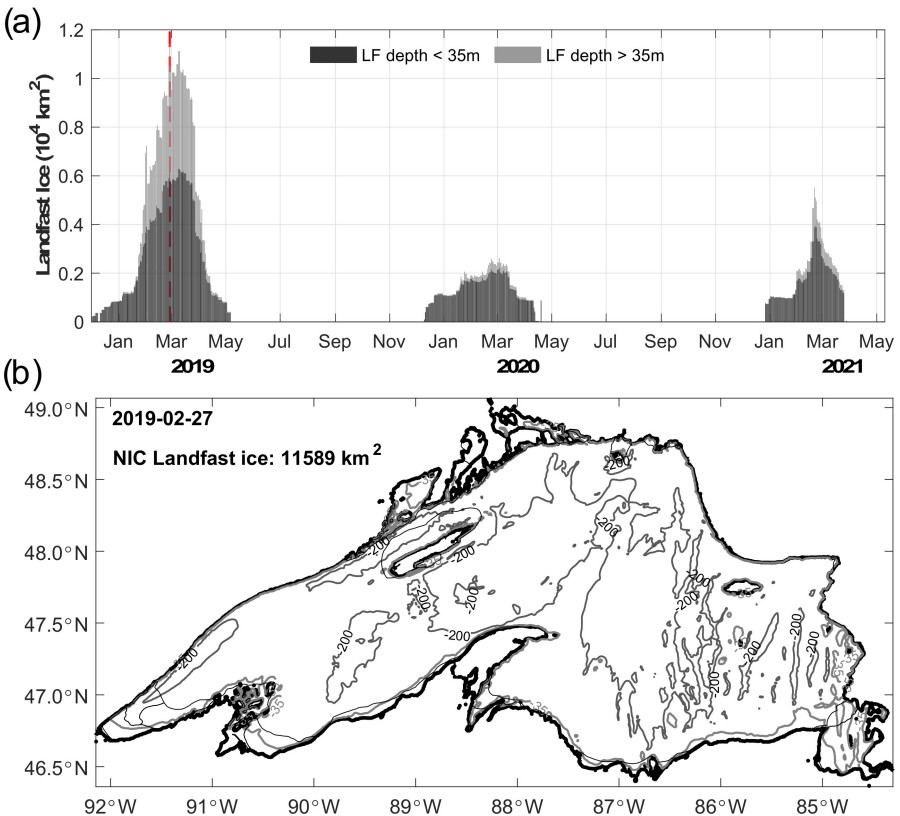

**Figure 2.** (**a**) Bar chart of landfast ice area (km$^2$) from the National Ice Center (NIC) between December 2018 to May 2021. Black indicates water depth less than 35 m and gray indicates greater than 35 m. The red vertical dashed line indicates the date of 27 February 2019 for (**b**). (**b**) A snapshot of landfast ice (color regions) from NIC on 27 February 2019. The first and the second gray contours indicate the 35 m and the 200 m isobath, respectively.

### 3.2. Model Simulation and Landfast Ice Frequency

Figure 3 shows snapshots of Lake Superior ice cover in each year from the NIC and the control case using the coupled FVCOM and UG-CICE models. The dates are different in each year and are selected prior to the maximum ice cover to avoid when Lake Superior is fully ice covered (100%). The percentages of ice cover (from 10% to greater than 90%) and landfast ice provided in the NIC are represented by the corresponding SIGRID codes. Note that ice cover of 10% is rarely present in the NIC analysis. In other words, the ice concentrations in the NIC generally start from 20%. In Figure 3a, landfast ice, which is marked as gray, is mainly located along coastal regions and tends to reach deep regions when ice coverage is high.

The same condition of lower than 10% ice concentration is applied to the model results when plotting Figure 3. As mentioned above, these three years from 2019 to 2021 include both high and low ice cover winters in Lake Superior. In general, the couped FVCOM and UG-CICE model is capable of reproducing the high and low ice cover years, as well as spatial coverage across different years (Figure 3). In 2018/19 (Figure 3b), the high ice cover year, Lake Superior is gradually covered by ice from the west and is entirely covered (the maximum ice cover) by high concentration ice on 9 March (not shown). Figure 3a,b show ice concentration a few days before the seasonal maximum ice cover date to display the similarity between observations and the model. In 2019/20 and 2020/21, the two low ice cover years, the model also reproduces the ice coverage near the coast and in western Lake Superior.

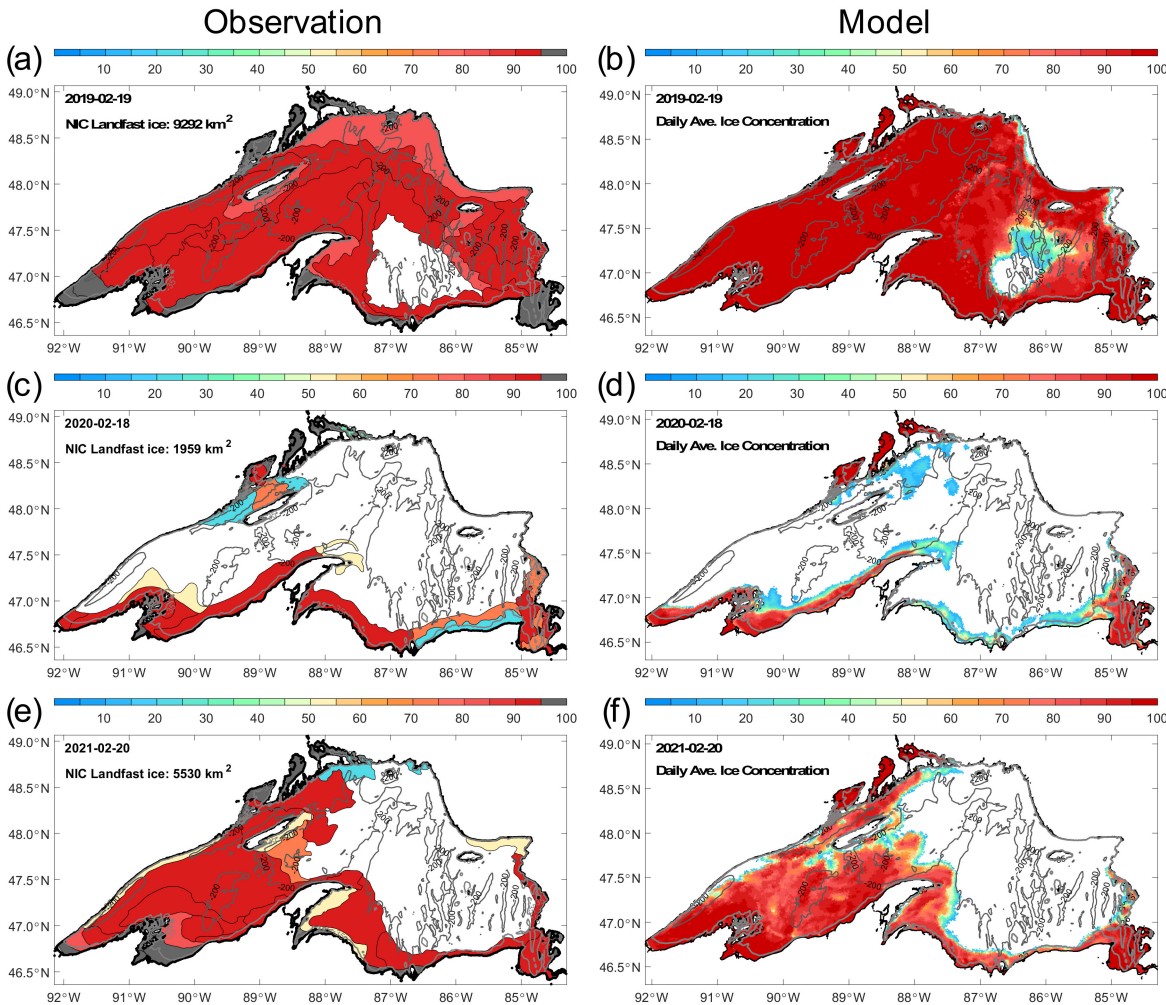

**Figure 3.** Snapshots of ice concentration from NIC observations (**a**,**c**,**e**) and model outputs (**b**,**d**,**f**) before the maximum ice cover of each year. Landfast ice from the NIC is marked as gray. The first and the second gray contours indicate the 35 m and the 200 m isobath, respectively.

In order to determine the hot spots in Lake Superior, Figure 4a shows the landfast ice frequency from the gridded NIC data for the period from January to April across all years. The months are selected by considering the timing of the growth and melt of the major landfast ice in Lake Superior. Landfast ice is rare in December and May, as these months generally mark the beginning and end of winter, respectively. Landfast ice occurs more frequently in Thunder Bay, Black Bay, and Nipigon Bay in the north, Whitefish Bay in the southeast, and Chequamegon Bay in the southwest. In particular, the most frequent landfast ice, with values close to 1, is located at Black Bay and Nipigon Bay, the two northernmost bays, indicating the hot spots of reoccurrence and long-lasting fast ice each year in Lake Superior. The frequent occurrence of landfast ice cover over these areas is likely associated with the shallow water depth (~35 m) and the semi-enclosed environment in which ice cover is easily trapped.

Figure 4b–d show the sensitivity test results using the coupled FVCOM and UG-CICE models. Our experiments mainly focus on the sensitivity of modeled landfast ice to $k_1$, $k_2$, and $T$. As described in the methods section, modeled landfast ice is determined based on three criteria, ice concentration (>90%), water depth (<35 m), and drifting speed (<0.4 cm s$^{-1}$), which is not exactly the same as the NIC landfast ice. Therefore, it is natural to expect differences between the model and the NIC observations, especially over the area of greater water depth.

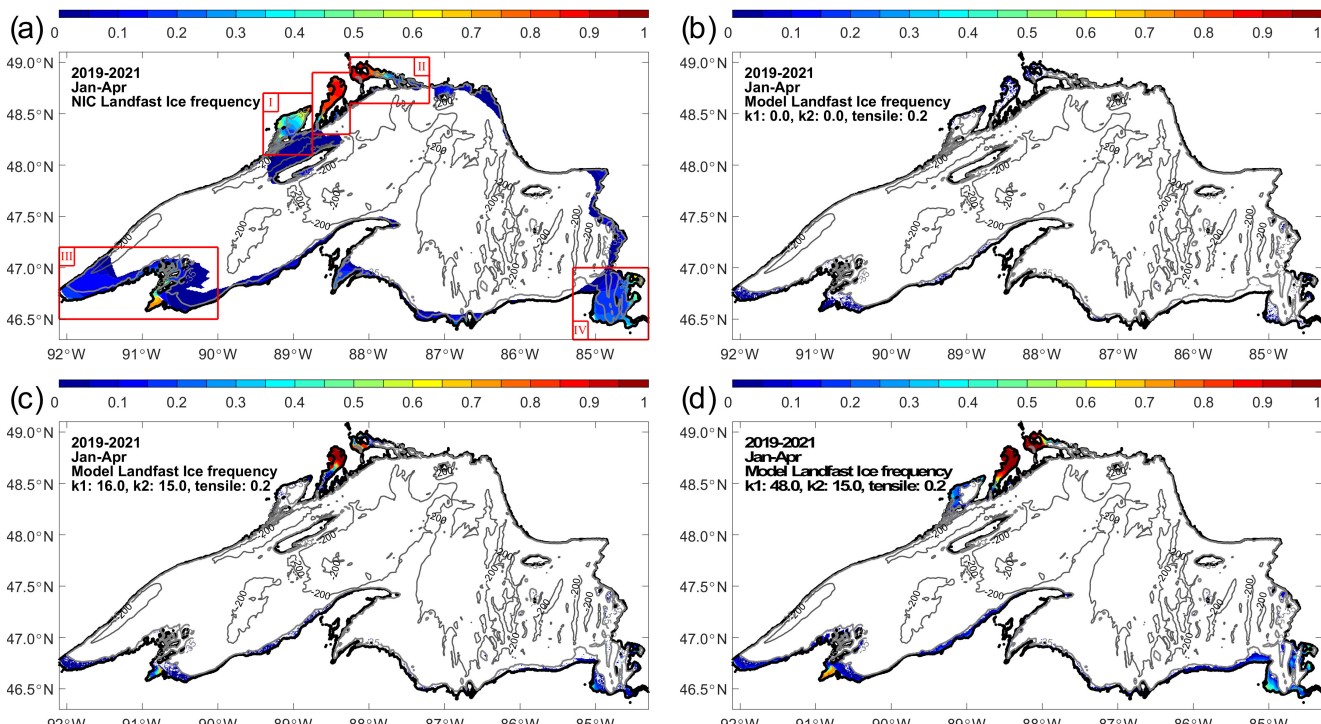

**Figure 4.** (**a**) Landfast ice frequency from the NIC in January to April from 2019 to 2021. (**b–d**) Same as Figure 4a but for model cases of no-basal stress (**b**), $k_1 = 16$ & $k_2 = 15\,\mathrm{Nm^{-3}}$ (**c**), and $k_1 = 48$ & $k_2 = 15\,\mathrm{Nm^{-3}}$ (**d**). The tensile strength is equal to 0.2 for the two cases with basal stress (**c,d**). The first and the second gray contours indicate the 35 m and the 200 m isobath, respectively. The 4 rectangles (domains I, II, III, & IV) indicate the sub-domains for comparing the landfast ice of the sensitivity experiments. Note that the domain II covers two bays in the north.

The landfast ice frequency in three selected cases, the control case (no-basal stress), $k_1 = 16$, and $k_1 = 48$, are shown in Figure 4b–d, respectively. For the two $k_1$ cases, $k_2$ is set to $15\,\mathrm{Nm^{-3}}$. The time period for all the cases (Figure 4b–d) and the NIC (Figure 4a) are the same, which covers January to April in 2019 to 2021. The control case (Figure 4b) reveals very low landfast ice frequency around the coastline of Lake Superior as expected. It is obvious that the inclusion of the basal stress parameterization improved the representation of landfast ice in the shallow regions (Figure 4c,d). Figure 4c,d show the effects of changing $k_1$ on landfast ice. When $k_1 = 16$, landfast ice mainly forms inside Black Bay and Nipigon Bay, the two bays in the north, Chequamegon Bay in the southwest, and surrounding Whitefish Bay in the southeast of Lake Superior. When $k_1 = 48$, the pattern of landfast ice frequency is similar to the case of $k_1 = 16$ but with higher frequencies in many areas. We did not see a notable sensitivity to the isotropic tensile strength parameter $T$ in the frequency spatial patterns (not shown).

To compare variations in landfast ice during the study period, we focused on the four regions shown in Figure 4a, where the landfast ice frequencies are high. Figures 5 and 6 show comparisons of the model results with perturbed $k_1$ and $k_2$ against the NIC, respectively, of the 4 regions (domains I, II, III, & IV) shown in Figure 4a. Note that Black Bay and Nipigon Bay are considered a group since Black Bay is easily covered by landfast ice over the entire bay. The observed landfast ice (Figures 5 and 6) is plotted for the total landfast ice area obtained from NIC (gray line) and the landfast ice area with water depths shallower than 35 m (black line). The total landfast ice area is still much higher than the landfast ice area at depths shallower than 35 m, especially in 2018/19. Note that domain II has the smallest difference in 2018/19, compared to the other domains. The comparisons between observations and the model will focus on the landfast ice at water depth shallower than 35 m.

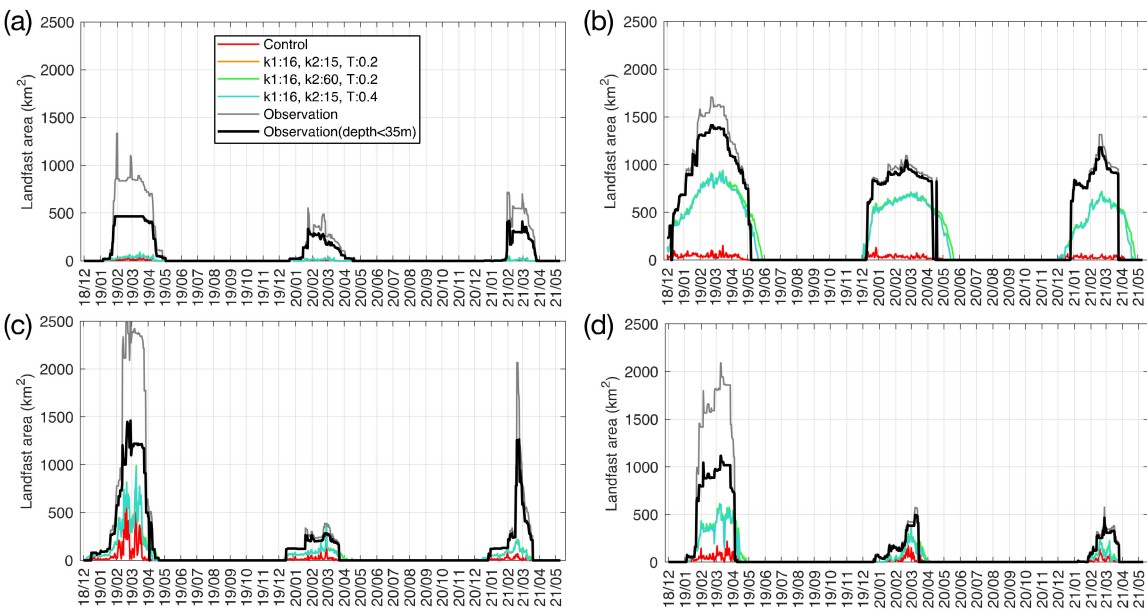

**Figure 5.** Time-series of the landfast ice area from model results and NIC observations for of the 4 domains in Figure 4a. The panels (**a–d**) correspond to these domains. The model output presents the results of the sensitivity experiments with $k_1 = 16$, $k_2$ values varying from 16 to 60 Nm$^{-3}$, and tensile strength (*T*) varying from 0.2 to 0.4. The gray line indicates the total landfast ice area obtained from NIC observations and the black line indicates the landfast ice area limited to water depths shallower than 35 m.

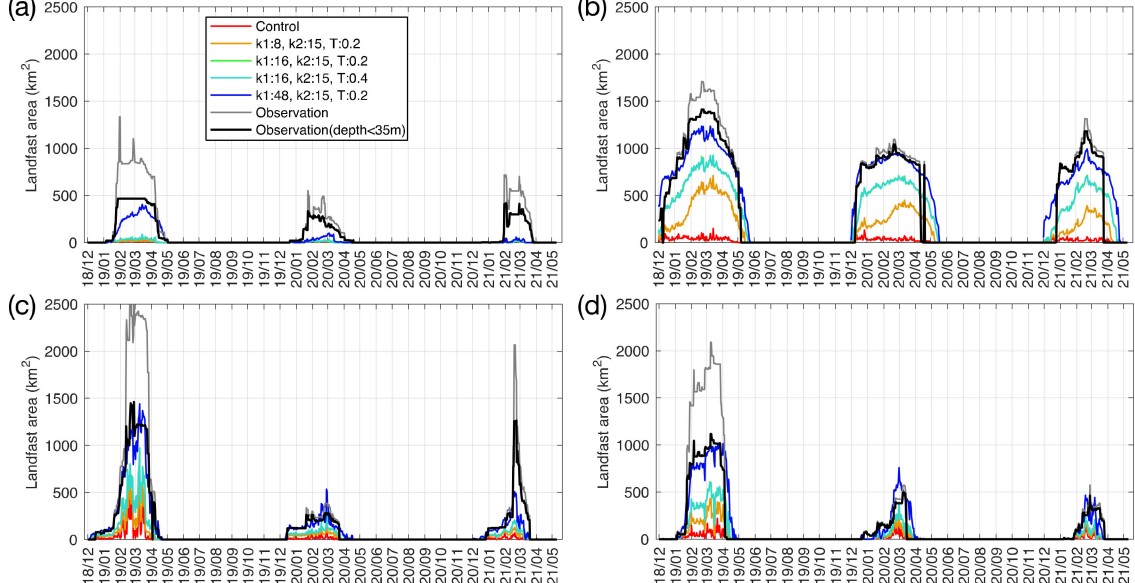

**Figure 6.** Same as Figure 5 for the time-series of the landfast ice area from model results and NIC observations for of the 4 domains in Figure 4a but with a fixed $k_2$ value of 15 Nm$^{-3}$, and $k_1$ varies from 8, 15, and 48, and tensile strength (*T*) varies from 0.2 to 0.4. The panels (**a–d**) correspond to these domains. The gray line indicates the total landfast ice area obtained from NIC observations and the black line indicates the landfast ice area limited to water depths shallower than 35 m.

In Figure 5, $k_1$ is fixed to 16, $k_2$ varies from 15 to 60 Nm$^{-3}$, and the isotropic tensile strength parameter *T* changes from 0.2 to 0.4. In all four regions, modeled landfast ice extent is improved when the parameterizations of the basal stress and tensile strength are included. The simulated landfast ice area changes very little when $k_1$ is fixed and the other

two parameters are changed. The three cases of $k_1 = 16$ are almost overlapping with each other, indicating that changing $k_2$ or $T$ does not have a large effect on the formation of landfast ice. Figure 5 indicates that the model has the best performance on landfast ice over domain II (Figure 5b), Black Bay and Nipigon Bay, compared to the other domains. In domain II, the three cases of $k_1 = 16$ reach about 50% and 60% of the total landfast ice and the landfast ice shallower than 35 m, respectively, in 2018/19. The ratios raise to over 75% for the years of 2019/20 and 2020/21. The high production of landfast ice in domain III also indicates that the model successfully simulates the initiation of landfast ice or lake ice in Lake Superior.

The model experiment with $k_1 = 16$ does not generate enough landfast ice in the other three domains compared to the NIC, especially in domains I and III. For these two domains, one reason for the low landfast ice in the model simulations could be because of the landfast ice threshold of 35 m water depth. A large portion of the NIC landfast ice outgrows the 35 m water depth, but the landfast ice of the model is restricted. This situation is improved when the criterion of a 35 m water depth is lifted. In domain IV, the cases with $k_1 = 16$ seem to generate landfast ice that is about half or more than half of the observations for water depths shallower than 35 m. The control case, in which the basal stress is not applied, generates very little landfast ice throughout the simulation period.

Figure 6 shows the time-series of landfast ice area for the sensitivity experiments with a fixed $k_2 = 15\,\mathrm{Nm}^{-3}$, $k_1 = 8$, 16, and 48 $\mathrm{Nm}^{-3}$, and the isotropic tensile strength parameter $T = 0.2$ and 0.4. The time-series of landfast ice is significantly affected when $k_1$ changes from 8 to 48 over domains II, III, and IV, indicating the major contribution of $k_1$ to landfast ice in Lake Superior. The landfast ice area when $k_1 = 48$ is the case that is the closest to the NIC throughout the study period, especially in the domains II, III, and IV during 2018/19. More landfast ice is generated as the $k_1$ increases due to the reduction in the critical mean ice thickness $h_c$. However, $k_1 = 48$ is much larger than the values tested in Lemieux et al. [20]. With $k_1 = 48$ and ice concentration $A = 1$, the critical ice thickness $h_c$ is about 0.2 m and 0.6 m at water depths of 10 m and 30 m, respectively. It could be unrealistic for ice with mean thicknesses of 0.2 m and 0.6 m to have a keel(s) that are 50 times larger that can ground at 10 m and 30 m depths, respectively. Further analyses may be needed in the future to inform whether this value is valid in Lake Superior; however, this is not within the scope of this study. Compared to $k_1$ and $k_2$, the tensile strength has very little effect on modeled landfast ice (Figures 5 and 6).

The influences of $k_1$, $k_2$, and $T$ on the formation of landfast ice may be attributed to Equation (1) and the mechanism of grounding. Equation (1) and the critical mean thickness $h_c$ reveal that $k_1$ is expected to have a larger impact on the formation of landfast ice compared to $k_2$ within a certain range. The limited effect of the tensile strength $T$ indicates that the grounding of fast ice is the dominant mechanism in Lake Superior. In the Arctic, the tensile strength $T$ has been shown to have different impacts in the Kara Sea, the East Siberian Sea, and the Laptev Sea [21]. Overall, the contributions of $k_1$, $k_2$, and $T$ on the formation of landfast ice are similar to the findings of Lemieux et al. [20,21]. Other aspects, such as ice formation in freshwater or lake conditions, could also potentially affect the simulation of landfast ice.

## 4. Summary and Conclusions

In this study, we evaluated landfast ice in Lake Superior, the largest freshwater lake on Earth by surface area, through the examination of ice analyses from the NIC, the addition of parameterizations for basal stress and ice tensile strength to the unstructured FVCOM-CICE hydrodynamic ice model, and by conducting several numerical experiments. This is the first study where the spatial and seasonal evolution of landfast ice are examined in a large freshwater lake using an unstructured-grid hydrodynamic ice model.

The observed landfast ice from the NIC from December 2018 to May 2021 was analyzed to understand the variation and frequency near the coast. The frequency of the observed landfast ice during the study period shows that landfast ice is initiated from the bays in the

northern region of the lake in December. Compared to the landfast ice extent estimated using the same criteria used in the previous studies for the Arctic Ocean, the NIC landfast ice extent in Lake Superior tends to be larger than modeled conditions, particularly in the anomalously cold winters with high ice coverage. This is because the NIC landfast ice often extends to the deeper areas (greater than 35 m water depth), which are filtered out in the criteria used for the model simulations. We note the limitations of landfast ice data reported by the NIC. First, satellite measurements that are used in the NIC data do not directly provide information on ice drifting speed, which is critical to determine whether ice is immobile or mobile. Second, satellite measurements are often limited by cloud cover and spatial footprint for visible sensors and SARs, respectively. Uncertainties associated with these limitations can be greater in deeper areas as the grounding of an ice keel(s) cannot play a role as a mechanism to make ice landfast.

The coupled FVCOM and UG-CICE models successfully simulated landfast ice extent in Lake Superior by implementing the basal stress parameterization and the ice tensile strength to the momentum equation [20,21]. We performed a sensitivity experiment to examine the major free parameters, $k_1$, $k_2$, and $T$. We found that the simulated landfast ice in Lake Superior is mainly controlled by $k_1$, the free parameter that determines the critical ice thickness $h_c$, and is less sensitive to the other two parameters. The greater influence of $k_1$ over $k_2$ can be attributed to the formula for basal stress, and the limited effect of tensile stress indicates the dominant mechanism of grounding [21]. The simulated landfast ice area is only close to the observed landfast ice with the same water depth criterion, e.g., less than 35 m, and when $k_1 = 48$, largely reducing the critical mean ice thickness. $k_1 = 48$ is higher than the values used in the previous model simulations of the Arctic Ocean [20,21]. Given that there are a few major differences between the previous studies and our application, including a large lake environment and the much higher spatial resolution (~200 m), whether the high $k_1$ value is adequate or not in the current application might warrant a future analytical study.

Our study demonstrates the validity of the existing landfast ice parameterizations of basal stress and ice tensile strength in large lake applications. The findings of this study have the potential to advance the forecasting capability of Great Lakes ice cover through implementation into NOAA's Great Lakes Operational Forecast System (GLOFS). Improved representation of landfast ice in the models will greatly increase the granularity of information on nearshore hydrodynamics and ice, and will therefore benefit decision-making related to navigation, shipping, and icebreaking operations.

**Author Contributions:** The paper was conceived of, written and revised by Y.L., A.F.-M. and E.J.A. and Y.L. and A.F.-M. carried out the model simulations and conducted the pre-processing and post-processing of results and observational data. All authors have read and agreed to the published version of the manuscript.

**Funding:** Funding was awarded to the Cooperative Institute for Great Lakes Research (CIGLR) through the NOAA Cooperative Agreement with the University of Michigan (NA17OAR4320152). This CIGLR contribution number is 1196.

**Institutional Review Board Statement:** Not applicable.

**Informed Consent Statement:** Not applicable.

**Data Availability Statement:** The daily shapefiles and gridded ascii files of Lake Superior can be accessed at the U.S. National Ice Center (http://usicecenter.gov/Products/GreatLakesHome, accessed on 5 June 2022).

**Acknowledgments:** This research was carried out with support of the National Oceanic and Atmospheric Administration (NOAA) Great Lakes Environmental Research Laboratory (GLERL). We greatly thank the U.S. National Ice Center (NIC) for providing the daily gridded Great Lakes ice products of shapefiles and ascii data (https://usicecenter.gov/Products/GreatLakesHome, accessed on 5 June 2022). We are grateful to three anonymous reviewers for their construction comments that helped to im-prove the manuscript significantly.

**Conflicts of Interest:** The authors declare no conflict of interest.

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
