# Peer review of "Simulating Landfast Ice in Lake Superior"

_jmse, doi:10.3390/jmse10070932_

Round 1

Reviewer 1 Report

This paper presents the improvement of the unstructured-grid hydrodynamic-ice model using the case of Lake Superior. Unfortunately, the methods described are unclear to me as soon as I am not a specialist in modeling. Nevertheless, as far as I understood, the research is appropriately conducted but requires some improvements.

First, the description of methods needs to be more extensive. This chapter does not contain a list of input components of the model as well as the model's output. This makes the purposes of modeling unclear.

Second, the criterion of 35 m is unclear. This depth is taken from the Arctic conditions, but what about mid-latitudes? Great Lakes, Caspian Sea, Gulf of St. Lawrence, Baltic Sea? Is this criterion universal for seasons with different severity? I see that the authors discuss this lower in the text, but it would be fair to estimate the accuracy of this criterion.

Third, the authors are not familiar with

Grass J.D. Ice scour and ice ridging studies in Lake Erie // Proceedings of the 7th International Symposium on Ice. Association of Hydraulic Engineering and Research – Hamburg, Germany, 1984. P. 33-43. 

Daly S.F. Characterization of the Lake Erie Ice Cover. U.S. Army Engineer Research and Development Center (ERDC), Cold Regions Research and Engineering Laboratory (CRREL), Hanover, USA. 2016. 100 p.

Li et al., 2020. Spatial and Temporal Variations in the Extent and Thickness of Arctic Landfast Ice. Remote Sensing, 12, 64; doi:10.3390/rs12010064

Also, I have some specific comments.

1. Line 17 – I found nothing about grounded keels in the results

2. Line 97 – The use of 35 m isobaths must be explained? It's more reasoned to show 50 m. Why these two isobaths? 100 m, 150 m? It is not correct from the point of view of cartography.

3. Line 99 – What is the source of bathymetry? Reference?

4. Line 159 – Why do you use these values? 48 & 60 are much higher than the above mentioned. Why 48 but not 32 or any other?

5. Line 164 – Using the 35 m criterion must be explained on a larger scale in text.

6. Lines 226-228 It is obvious that the coverage of landfast ice determined by NIC extended toward depths greater than 35m. The landfast ice reaches the area beyond 200 m depth – How do you consider this in calculations? 

7. Line 370 – Check the limits of ice keel depths in the Great Lakes in Daly, 2016; Grass, 1984.

Author Response

This paper presents the improvement of the unstructured-grid hydrodynamic-ice model using the case of Lake Superior. Unfortunately, the methods described are unclear to me as soon as I am not a specialist in modeling. Nevertheless, as far as I understood, the research is appropriately conducted but requires some improvements.

Response:
We appreciate the reviewer’s suggestions and comments. We have revised the manuscript substantially and the manuscript has also been checked by a native English-speaking colleague.

First, the description of methods needs to be more extensive. This chapter does not contain a list of input components of the model as well as the model's output. This makes the purposes of modeling unclear.

Response:

The purpose of this study is to address the gaps of landfast ice parameterizations in the FVCOM (Chen et al., 2006) (see the last paragraph of the Introduction). The FVCOM simulates hydrodynamics in the ocean, lakes, rivers, and bays. In other words, it simulates behavior of water flows and thermal changes with realistic initial and boundary conditions, including the bathymetry, the water body, and the forcing of the wind. These conditions are described in the first and second paragraphs of Section 2 Materials and Methods. In these two paragraphs, the general configurations of the model, the type of input wind forcing, and the coupled ice model (CICE) are adequately described. In addition, the FVCOM and the coupled version with CICE have been wildly applied to the research of the Great Lakes (Gao et al., 2011; Bai et al., 2013; Luo et al., 2015; Niu et al., 2015; Anderson et al., 2018; Bai et al., 2020; Fujisaki-Manome et al., 2020a, 2020b). The thorough details of the model configuration are listed in these papers and we believe referring to them would be sufficient, given they are not foci of this study. The third to the fifth paragraphs describe the parameterizations of basal stress and tensile strength following Lemieux et al [2015, 2016], which is newly included in the FVCOM-CICE models in this study. The following paragraph describes the parameterizations that are examined in this study. As mentioned in the second paragraph, these parameterizations were not incorporated with the FVCOM. We believe these descriptions are sufficient for the purpose of modeling.

Second, the criterion of 35 m is unclear. This depth is taken from the Arctic conditions, but what about mid-latitudes? Great Lakes, Caspian Sea, Gulf of St. Lawrence, Baltic Sea? Is this criterion universal for seasons with different severity? I see that the authors discuss this lower in the text, but it would be fair to estimate the accuracy of this criterion.
Response:

The criteria for determining landfast ice are defined differently in previous studies. Johnson et al. (2012) and Rozman et al. (2011) applied a mask to set the ice velocity to zero; Lieser et al. (2004) considered the criteria of sea ice thickness over a certain water depth; Wang et al. (2014) set the criteria of sea ice velocity over a certain depth. Lemieux et al. (2015, 2016) applied the criteria of Wang et al. (2014), which is criterion of 35 m, to their study for determining landfast ice. Since we are using the same parameterization to Lemieux et al. [2015, 2016], it is reasonable to applying similar criterion to determine landfast ice. It is unfortunate that there are not enough observational studies to validate the accuracy of the criterion of 35 m depth in Lake Superior. The reason of this criterion of 35 m is described in the text.

The criterion is not universal. The following studies show the depth varieties of landfast ice over the world. Mahoney et al. (2007) reported that landfast ice of the Beaufort Sea is confirmed within the 20-m isobath. Wang et al. (2010) mentioned that landfast ice of Lake Erie is attached or formed on the shallower shore around 5-10 m. Itkin et al. (2015) focused on the landfast ice regions shallower than 25m over the Arctic. In addition, Liu et al. (2022) applied the method of Lemieux et al. [2015] to the area of the Totten Glacier, East Antarctica with criterion on ice drifting speed since the depth of East Antarctica is much deeper than coastal of the Arctic.

Third, the authors are not familiar with

Grass J.D. Ice scour and ice ridging studies in Lake Erie // Proceedings of the 7th International Symposium on Ice. Association of Hydraulic Engineering and Research – Hamburg, Germany, 1984. P. 33-43. 

Daly S.F. Characterization of the Lake Erie Ice Cover. U.S. Army Engineer Research and Development Center (ERDC), Cold Regions Research and Engineering Laboratory (CRREL), Hanover, USA. 2016. 100 p.

Li et al., 2020. Spatial and Temporal Variations in the Extent and Thickness of Arctic Landfast Ice. Remote Sensing, 12, 64; doi:10.3390/rs12010064

Response:
The first two references, Grass (1984) and Daly (2019), seem to be inappropriate and unrelated to the current study. Both of them are focused on the ice-ridge in Lake Erie. Daly (2019) reported that the maximum keel depth of ice-ridge is around 10-25 m depth in Lake Erie. Although the ice-ridge is relevant to the total depth and grounding process, it is hard to link to landfast ice in Lake Superior.
Li et al. (2020) is added to the first paragraph of the Introduction to support the background of landfast ice in the Arctic.

Also, I have some specific comments.

  1. Line 17 – I found nothing about grounded keels in the results
    Response: This sentence is rewritten for a clear description. The basal stress parameterization is to represent the effects of grounded ice keels and tensile stress of ice. We did not directly simulate the grounded ice keels.
  2. Line 97 – The use of 35 m isobaths must be explained? It's more reasoned to show 50 m. Why these two isobaths? 100 m, 150 m? It is not correct from the point of view of cartography.
    Response: These two isobaths, 35 m and 200 m, are placed with spatial purpose.
    The 35 m isobath represents the criterion of depth for determining the landfast ice in Lake Superior. The 200 m isobath represents the deeper regions of Lake Superior, which is harder for landfast ice to ground. It is not necessary to mark isobaths simply following the cartography with no purpose. For instance, a recently published paper on landfast ice (Figure 9 of Liu et al., 2022) marks the isobaths of 25 m and 60 m based on their needs.
  3. Line 99 – What is the source of bathymetry? Reference?
    Response: The bathymetry is plotted from the model topography. The source of the bathymetry from “3 arc-second bathymetry data from the NOAA National Centers for Environmental In-formation (NCEI)”, which is included in the first paragraph of Section 2.
  4. Line 159 – Why do you use these values? 48 & 60 are much higher than the above mentioned. Why 48 but not 32 or any other?
    Response: According to Lemieux et al. [2015], these values should be optimized through numerical experiments, and we had tested several of them. The results shown here is to reveals the effects of k1 and k2 with extremely high value (k1 = 48 & k2 = 60 Nm-3). As shown in Fig. 5, the k2 = 60 Nm-3 does not shows visible difference from k2 = 15 Nm-3, indicating that k2 has little effect when varying from 15 to 60 Nm-3.
  5. Line 164 – Using the 35 m criterion must be explained on a larger scale in text.
    Response: The reason of chosen the criterion of 35 m is expressed in the previous comment and it is now included in the text.
  6. Lines 226-228 It is obvious that the coverage of landfast ice determined by NIC extended toward depths greater than 35m. The landfast ice reaches the area beyond 200 m depth – How do you consider this in calculations? 
    Response: The general definition for landfast ice is that sea ice is fastened to the coastline, to the sea floor along shoals, or to grounded iceberg. It is possible for landfast ice to reach deeper water regions but it is rare according to the landfast ice frequency in Figure 4a. Observation shows that landfast ice tends to reappear at regions with shallower water depths in Lake Superior. This situation is not included in the current method but it can be considered by enlarging the criterion of depth. However, enlarging the criterion of depth leads to more modeled landfast ice in the middle of the lake compared to the observation, which is unrealistic.
  7. Line 370 – Check the limits of ice keel depths in the Great Lakes in Daly, 2016; Grass, 1984.
    Response: The sentences here are mainly to describe the relationship between k1 and critical ice thickness based on equation 1, which was proposed by Lemieux et al. [2015]. It is not the actual ice keel depths that are simulated. Critical ice mean thickness hc = Ahw/k1 is described in Section 2. Given too large k1 (here is 48), the same magnitude of critical ice thickness (here are 0.2 m and 0.6 m) will result in much deep water depths hw and that is the unrealistic part.
    For comparison, Lemieux et al. [2015] mentioned that k1 = 8 presents the best landfast ice results in the Arctic Ocean. With ice concentration A = 1 and mean ice thickness ~ 2 m (based on Fig. 9a of Lemieux et al. [2015]), the water depth hw is about 16m.

Reference:

Liu, Y., Losch, M., Hutter, N., and Mu, L., 2022, A New Parameterization of Coastal Drag to Simulate Landfast ice in Deep Marginal Seas in the Arctic, Journal of Geophysical Research: Oceans, 127, e2022JC018413. https://doi. org/10.1029/2022JC018413.

Reviewer 2 Report

In this study, the authors implemented the basal stress and the ice tensile strength parameterizations to the coupled FVCOM and UG-CICE models, and demonstrate the impacts of three free parameters on predicting the seasonal evolutions of landfast ice production/extent in Lake Superior. This paper is motivated well in that sense. A notable conclusion is that the updated FVCOM-UG-CICE is indeed valid in large-lake landfast ice prediction, and the critical thickness parameter k1 exerts major contributions in determining the model performance in simulating landfast ice production. This work is valuable in advancing the nearshore ice and hydrodynamic modeling among Laurentian Great Lakes. Thus, I recommend the manuscript be accepted for publication after a revision that can be mostly considered to be minor, with a focus on improving the presentation. I provide more detailed comments below.

The grammar is not accurate nor concise (unnecessary words). Some comments are presented below and I hope the authors make another effort to improve the writing.

1. There should be a space between the values and the unit, i.e., L96 “200m”, L227 “35m”.

2. L456-466 vs L467-468: The reference format should be unified with full name or abbreviation of a journal.

3. L189: The full name is required when the abbreviation is given for the first time.

4. L286-288: It is unnecessary to mention again the criterion for detecting landfast ice based on model simulation.

5. L365-366: The tense of the main and subordinate clauses should be consistent.

6. As you mentioned, more landfast ice is generated as the k1 increases due to the reduction of the critical mean ice thickness hc. I don’t quite understand how it works and why the maximum basal stress parameter k2 and the isotropic tensile strength parameter T show little impacts. Maybe a detailed explanation about how the three parameters influence the landfast ice prediction is needed for better presentation.

7. The tables and figures are in general with good quality. Here are a few points that I noticed:

 Figure 1: It’s better to include unstructured triangular mesh used in FVCOM-UG-CICE.

Figure 2: Suggest to choose another color to make the vertical dashed line in Figure 2a clear, and to replace “Fig.1b” with “Fig.2b” in L 242.

Figure 3: I wonder why different color bars are used for observations and simulations.

Figure 4: Replace “k2” with “k2”.

Figure 6: Replace “k1” with “k1”.

Author Response

In this study, the authors implemented the basal stress and the ice tensile strength parameterizations to the coupled FVCOM and UG-CICE models, and demonstrate the impacts of three free parameters on predicting the seasonal evolutions of landfast ice production/extent in Lake Superior. This paper is motivated well in that sense. A notable conclusion is that the updated FVCOM-UG-CICE is indeed valid in large-lake landfast ice prediction, and the critical thickness parameter k1 exerts major contributions in determining the model performance in simulating landfast ice production. This work is valuable in advancing the nearshore ice and hydrodynamic modeling among Laurentian Great Lakes. Thus, I recommend the manuscript be accepted for publication after a revision that can be mostly considered to be minor, with a focus on improving the presentation. I provide more detailed comments below.

Response:
We appreciate the reviewer’s suggestions and comments on the grammar errors. We have revised the manuscript substantially and the manuscript has also been checked by a native English-speaking colleague.

The grammar is not accurate nor concise (unnecessary words). Some comments are presented below and I hope the authors make another effort to improve the writing.

  1. There should be a space between the values and the unit, i.e., L96 “200m”, L227 “35m”.
    Response: Corrected.
  2. L456-466 vs L467-468: The reference format should be unified with full name or abbreviation of a journal. 
    Response: The format of reference is modified to fit the JMSE standard.
  3. L189: The full name is required when the abbreviation is given for the first time.
    Response: Corrected.
  4. L286-288: It is unnecessary to mention again the criterion for detecting landfast ice based on model simulation. 
    Response: We think it is worth to list the criteria for determining the modeled landfast ice since the criteria is described in a large paragraph. This sentence is modified to be a reminder for the cause of differences on landfast ice between the observation and the model instead of a repeated list.
  5. L365-366: The tense of the main and subordinate clauses should be consistent.

Response: Corrected.

  1. As you mentioned, more landfast ice is generated as the k1 increases due to the reduction of the critical mean ice thickness hc. I don’t quite understand how it works and why the maximum basal stress parameter k2 and the isotropic tensile strength parameter T show little impacts. Maybe a detailed explanation about how the three parameters influence the landfast ice prediction is needed for better presentation.
    Response: According to eq. 1 and the definition of critical mean ice thickness, both k1 and k2 are contributing to the basal stress, which leads to the formation of landfast ice. However, the kc is in inverse ratio to k1 and k2 is a factor in eq. 1. As k1 increases, kc decreases, reducing the threshold of forming landfast ice. Mathematically, this makes k1 more effective in forming landfast ice than k2. The limited effect of the tensile strength T indicates that the grounding of fast ice is the dominant mechanism in Lake Superior. A new paragraph is added to the end of Section 3.2 to express the contributions of k1, k2, and tensile strength.
  2. The tables and figures are in general with good quality.
    Response: Thanks for the comment!

Here are a few points that I noticed:

Figure 1: It’s better to include unstructured triangular mesh used in FVCOM-UG-CICE.

Response: The unstructured triangular mesh of the FVCOM-UG-CICE is included as Fig. 1b.

Figure 2: Suggest to choose another color to make the vertical dashed line in Figure 2a clear, and to replace “Fig.1b” with “Fig.2b” in L 242.

Response: The figure is modified as suggested and the caption is corrected.

Figure 3: I wonder why different color bars are used for observations and simulations.

Response: The purpose of Fig. 3 is to compare the ice concentration between observation and model. The color bar with gray color in Fig.3a is to display the landfast ice, which is defined by NIC.

Figure 4: Replace “k2” with “k2”.

Response: Corrected.

Figure 6: Replace “k1” with “k1”.

Response: Corrected.

Reviewer 3 Report

The Authors simulated landfastice on Lake Superior. I believe that the work is valuable and does not require major corrections. It brings new information to the study of the studied problem. I only think that the number of cited publications could be greater.

Author Response

Response: Thanks for the comment!